# The latent tuberculosis cascade-of-care among people living with HIV: A systematic review and meta-analysis

Mayara Lisboa Bastos[1,2,3,4], Luca Melnychuk[3], Jonathon R. Campbell[1,3,4,5], Olivia Oxlade[4], Dick Menzies[1,3,4,5]*

1 Respiratory Epidemiology and Clinical Research Unit, Research Institute of the McGill University Health Centre, Montreal, Canada, 2 Social Medicine Institute, State University of Rio de Janeiro, Rio de Janeiro, Brazil, 3 Department of Medicine, McGill University, Montreal, Canada, 4 McGill International TB Centre, McGill University, Montreal, Canada, 5 Department of Epidemiology, Biostatistics & Occupational Health, McGill University, Montreal, Canada

* dick.menzies@mcgill.ca

## Abstract

**Data Availability Statement:** All relevant data are within the manuscript and its Supporting Information files.

### Background

Tuberculosis preventive therapy (TPT) reduces TB-related morbidity and mortality in people living with HIV (PLHIV). Cascade-of-care analyses help identify gaps and barriers in care and develop targeted solutions. A previous latent tuberculosis infection (LTBI) cascade-of-care analysis showed only 18% of persons in at-risk populations complete TPT, but a similar analysis for TPT among PLHIV has not been completed. We conducted a meta-analysis to provide this evidence.

### Methods and findings

We first screened potential articles from a LTBI cascade-of-care systematic review published in 2016. From this study, we included cohorts that reported a minimum of 25 PLHIV. To identify new cohorts, we used a similar search strategy restricted to PLHIV. The search was conducted in Medline, Embase, Health Star, and LILACS, from January 2014 to February 2021. Two authors independently screened titles and full text and assessed risk of bias using the Newcastle–Ottawa Scale for cohorts and Cochrane Risk of Bias for cluster randomized trials. We meta-analyzed the proportion of PLHIV completing each step of the LTBI cascade-of-care and estimated the cumulative proportion retained. These results were stratified based on cascades-of-care that used or did not use LTBI testing to determine eligibility for TPT. We also performed a narrative synthesis of enablers and barriers of the cascade-of-care identified at different steps of the cascade.

A total of 71 cohorts were included, and 70 were meta-analyzed, comprising 94,011 PLHIV. Among the PLHIV included, 35.3% (33,139/94,011) were from the Americas and 29.2% (27,460/94,011) from Africa. Overall, 49.9% (46,903/94,011) from low- and middle-income countries, median age was 38.0 [interquartile range (IQR) 34.0;43.6], and 65.9% (46,328/70,297) were men, 43.6% (29,629/67,947) were treated with antiretroviral therapy

**Funding:** This work was funded by the Bill & Melinda Gates Foundation (Grant Number INV-003634). The initial study questions for the papers included in the PLOS Collection were drafted together with input from staff of the Bill & Melinda Gates Foundation, but they had no further role in study design, data collection and analysis, decision to publish, or preparation of the manuscript.

**Competing interests:** The authors have declared that no competing interests exist.

**Abbreviations:** ART, antiretroviral therapy; IGRA, interferon gamma release assay; IQR, interquartile range; LTBI, latent tuberculosis infection; PEPFAR, President's Emergency Plan for AIDS Relief; PLHIV, people living with HIV; RCT, randomized clinical trial; TB, tuberculosis; TPT, tuberculosis preventive therapy; TST, tuberculin skin test.

(ART), and the median CD4 count was 390 cell/mm3 (IQR 312;458). Among the cohorts that did not use LTBI tests, the cumulative proportion of PLHIV starting and completing TPT were 40.9% (95% CI: 39.3% to 42.7%) and 33.2% (95% CI: 31.6% to 34.9%). Among cohorts that used LTBI tests, the cumulative proportions of PLHIV starting and completing TPT were 60.4% (95% CI: 58.1% to 62.6%) and 41.9% (95% CI:39.6% to 44.2%), respectively. Completion of TPT was not significantly different in high- compared to low- and middle-income countries. Regardless of LTBI test use, substantial losses in the cascade-of-care occurred before treatment initiation. The integration of HIV and TB care was considered an enabler of the cascade-of-care in multiple cohorts. Key limitations of this systematic review are the observational nature of the included studies, potential selection bias in the population selection, only 14 cohorts reported all steps of the cascade-of-care, and barriers/facilitators were not systematically reported in all cohorts.

## Conclusions

Although substantial losses were seen in multiple stages of the cascade-of-care, the cumulative proportion of PLHIV completing TPT was higher than previously reported among other at-risk populations. The use of LTBI testing in PLHIV in low- and middle-income countries was associated with higher proportion of the cohorts initiating TPT and with similar rates of completion of TPT.

## Author summary

### Why was this study done?

- Tuberculosis (TB) remains as one of the main causes of deaths among people living with HIV (PLHIV).

- Tuberculosis preventive therapy (TPT) reduces TB-related morbidity and mortality PLHIV.

- Previous meta-analysis has shown that many losses occurred in the TPT cascade-of-care. However, a similar analysis has not been conducted in PLHIV.

### What did the researchers do and find?

- We conducted a systematic review and meta-analysis evaluating the TPT cascade-of-care among PLHIV. We constructed 2 cascade-of-care frameworks: (1) studies that did not use LTBI tests to determinate TPT eligibility; and (2) studies that used LTBI tests to determinate TPT eligibility.

- We performed stratified analyses by income setting (high-income versus low- and middle-income countries) and type of clinics where patients were followed (HIV clinics versus other clinics). We also performed meta-regression using adjusting these 2 variables.

- Among the cohorts that did not use LTBI tests, the cumulative proportion of PLHIV completing TPT was 33.2% and 41.9% among cohorts that used LTBI tests. This was not statistically significant when we performed meta-regression by income and type of clinics.

**What do these findings mean?**

- The cumulative proportion of PLHIV completing TPT was higher than was previously reported among other at-risk populations.

- Recommendation and initiation of TPT was higher, and completion similar among cohorts that used LTBI tests, compared to cohorts offered TPT without LTBI testing.

- The use of LTBI test was not an important barrier for TPT.

- Substation losses remained in the TPT cascade-of-care, and continuous efforts are necessary to improve TPT care among PLHIV.

## Introduction

Tuberculosis (TB) remains a significant public health problem, particularly among people living with HIV (PLHIV). In 2019 alone, nearly 25% of PLHIV with TB disease died [1]. Tuberculosis preventive therapy (TPT) works synergistically with, and independently of, antiretroviral therapy (ART) to reduce TB incidence among PLHIV [2–4].

To scale up TPT in PLHIV, WHO has simplified its algorithm for TPT initiation by not requiring latent tuberculosis infection (LTBI) tests prior to initiation [4]. Either a tuberculin skin test (TST) or interferon gamma release assay (IGRA) can identify people who have LTBI, but these tests have reduced sensitivity among PLHIV due to impaired T-cell immunity. While PLHIV with a positive LTBI test are at substantially increased risk for active TB compared to PLHIV with a negative test, those with a negative test still experience TB disease at rates about 5 times higher than the general population [5]. For this reason, WHO recommendations permit TPT without the requirement of LTBI testing.

In 2019, 50% (3.5 million) of PLHIV newly enrolled in care initiated TPT compared to 1.5 million initiating TPT in 2018 [1]. However, these figures fail to capture the complete picture. Half of individuals eligible for TPT never initiated it, and it is uncertain how many of those initiated TPT completed it [1]. Thus, important barriers other than LTBI testing remain to be elucidated.

Cascade-of-care frameworks are increasingly used to identify gaps and barriers in care in order to develop targeted solutions [6–9]. These frameworks describe population-level engagement in the sequential steps of healthcare delivery systems in which patients must pass through multiple interventions to reach a desired outcome. Such cascades have been invaluable in highlighting gaps in HIV diagnosis and treatment implementation [10] and more recently have been used to broadly assess TPT uptake [11].

To help identify care gaps and potential targeted solutions, we conducted a systematic review and meta-analysis evaluating the LTBI cascade-of-care for TPT among PLHIV.

## Methods

### Objectives, search strategy, and selection criteria

Our systematic review and meta-analysis is reported according to PRISMA guidelines (S1 PRISMA Checklist) [12], and its protocol was registered in PROSPERO (CRD42020190264). The overall objective of this present systematic review was to quantify the cumulative

proportion of PLHIV completing each step of the LTBI cascade-of-care and to summarize health systems barriers and interventions to overcome those barriers identified for each step.

We first screened potential titles from a previous systematic review on the LTBI cascade-of-care published in 2016 [11]. This review had screened articles in 3 databases (Medline, Health Star, Embase) from 1946 to April 12, 2015, and it included different populations at risk of developing active TB, including PLHIV. For the identification of new cohorts, we updated the search, rerunning the search strategy using similar search terms, but with a focus in PLHIV (S1 Search Strategy) in the abovementioned databases, from January 1, 2014, to February 17, 2021. To expand our search to non-English publications, we searched one additional database, LILACS, from the inception date to February 17, 2021. For this database, we used a combination of English, Spanish, and Portuguese terms (strategy available in S1 Search Strategy). We also identified additional relevant articles from the reference list of the included studies and from another published systematic review [5].

Two reviewers (MLB and LM) independently screened titles, abstracts, and full text. When a consensus was not achieved, a third reviewer was consulted (DM).

Studies published in English, French, Portuguese, Spanish, and Chinese were eligible for inclusion. The studies had to report at least 2 consecutive steps of the cascade-of-care (defined in data extraction session and in Figs 1 and 2), have at least 25 PLHIV in the first step reported in that study, and report the use or not of LTBI tests (either TST or IGRA) to determine TPT eligibility. If the population was not exclusively PLHIV, the steps of the cascade-of-care had to be stratified by HIV status. We excluded studies which the objective was focused only on active TB case finding in PLHIV, and they did not investigate outcomes related to LTBI treatment. We excluded individual-level randomized clinical trials (RCTs) that evaluated efficacy of LTBI regimens. Editorials, opinion letters, and conference abstracts were also excluded.

## Data extraction

Two reviewers (MLB and LM) extracted 20% of the data using a standardized data form, then findings were checked for concordance. The agreement was high (95%), thus, a single reviewer (MLB) extracted the remaining data. Data extracted included study design, country, level of care (primary, secondary, or tertiary), type of service (TB, HIV, and other services), if an LTBI test was used or not, and the type of LTBI test used (IGRA or TST), if applicable. We collected information on the characteristics of the population including age, sex, ART, CD4 cells count, and LTBI regimen prescribed. We accepted the definition of a positive LTBI test (either IGRA or TST) as reported by the original studies. Within each cohort, we extracted the number of persons reaching each of the following steps of the cascade-of-care: (i) initially identified; (ii) tested for LTBI; (iii) LTBI test result available (TST read, or valid IGRA result received by providers); (iv) completed medical evaluation (including chest X-ray); (v) TPT recommended by providers; (vi) TPT accepted and started; and (vii) LTBI treatment completed (Figs 1 and 2). We considered TPT to have been recommended, if the study explicitly described a step as "providers recommendation," or if the study provided eligibility criteria for patients to receive TPT. In these studies, we assumed that the patients that met the center's eligibility criteria, they had received a provider recommendation for TPT. Finally, narrative comments related to barriers and enablers at each of these steps were collected from each study.

## Quality assessment

Two reviewers (MLB and LM) independently assessed risks of bias, and any disagreements were solved through consensus. For observational studies, we adapted the Newcastle–Ottawa Scale for cohorts [13], which included questions related to the ascertainment of exposure and

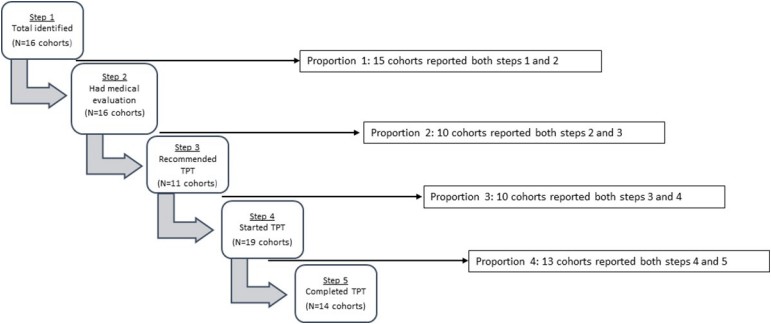

**Fig 1. Cascade framework used for analysis of cohorts that did not use LTBI tests (*N* = 21 cohorts).** LTBI, latent tuberculosis infection; TPT, tuberculosis preventive therapy.

the outcome assessments. We included an additional question related to population selection (S1 Table). For cluster randomized trials, we assessed the risk of bias using the most relevant questions from the Cochrane Risk of Bias tool [14] (S2 Table).

## Data analyses

For the quantitative analyses of the cascade-of-care, we considered 2 LTBI management approaches, following WHO algorithms, depending on whether or not programs used LTBI tests to guide treatment [4]. For the first approach, we restricted our analysis to cohorts **that did not use LTBI tests** to determine TPT eligibility, while in the second approach, we included only **cohorts that used LTBI tests** (either TST or IGRA). Figs 1 and 2 provide the framework for both approaches and the steps of the cascade-of-care that were analyzed within each.

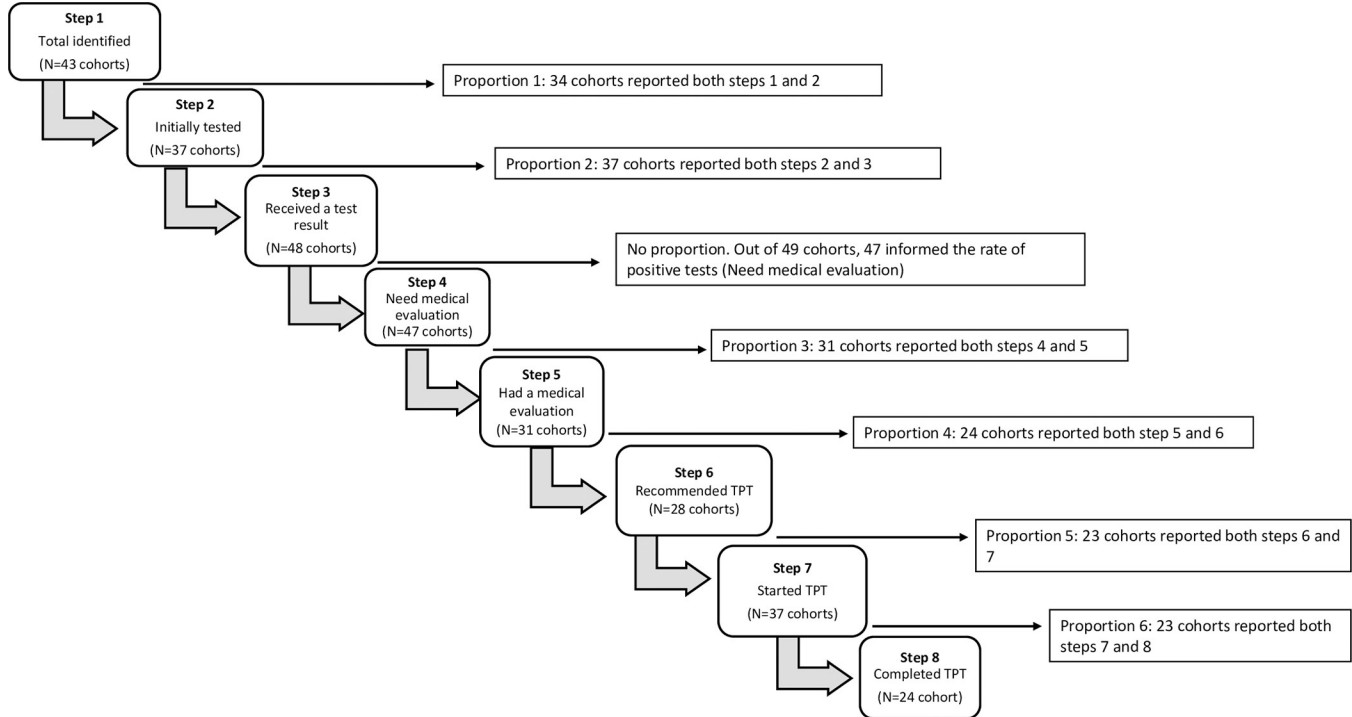

**Fig 2. Cascade framework for analysis of cohorts that used LTBI tests (*N* = 49 cohorts).** LTBI, latent tuberculosis infection; TPT, tuberculosis preventive therapy.

## Meta-analyses

To understand where the losses occurred in both cascade-of-care strategies, we meta-analyzed the proportions of PLHIV completing each step of the cascade-of-care. All proportions were meta-analyzed in R in the package meta (version 4.10–0) [15], using *metaprop* function. We meta-analyzed using generalized linear mixed models with fixed or random effects with a binomial distribution and logit link; pooled estimates were back transformed into proportions. The cumulative proportion retained in the cascade-of-care was estimated by multiplying the pooled estimated proportion completing each step by the pooled estimated proportion completing the preceding step. The same method was used for the confidence intervals, i.e., the inferior limit of each step was multiplied by the inferior limit of the preceding step, and the superior limit of each step was multiplied by the preceding superior limit. The choice of presenting our main analyses using fixed effect method was due to this method of calculating cascade confidence intervals for cumulative proportions and clearer visual presentation in graphic displays. However, all main meta-analyses using random effect models are presented in the supporting information tables (S1–S11 Tables).

To visually explore the variability of proportion within the cohorts, we generated forest plots of each proportion. As in our primary analyses, we stratified the forest plots by the use or not of LTBI tests.

## Stratified analyses

We performed 4 stratified meta-analyses: (1) stratified one the World Bank classification of the countries where the study was performed (high-income versus low- and middle-income) [16]; (2) stratified into cohorts followed in HIV clinics, or followed in any other type of clinic; (3) we restricted to only cohorts that reported data in all steps of the cascade-of-care; and (4) in cohorts that used LTBI tests, we stratified according to the type of LTBI tests used (TST versus IGRA).

## Meta-regression

We first meta-analyzed the number of PLHIV completing TPT divided by the number of PLHIV considered eligible for TPT (this was the total number identified if LTBI tests were not used, or the number of PLHIV identified multiplied by the prevalence of positive LTBI test in that cohort). A random effect meta-analysis was performed by fitting generalized linear mixed models with a binomial distribution and logit link; pooled estimates were back transformed into proportions. We used the package meta in R (version 4.10–0) [15], using *metaprop* function.

We then conducted 3 meta-regression models, each with one of the following 3 variables: (i) use of LTBI tests (used or not); (ii) income setting (high-income versus low- and middle-income); and (iii) type of service offering TPT (HIV-specific services versus other services). We interpreted the variable as significantly associated with TPT completion if the *p*-value was less than 0.05. We used the package meta in R (version 4.10–0) [15], using *metareg* function.

## Narrative synthesis

Finally, to understand why losses and retentions occurred in different steps of the cascade-of care, we extracted from the included papers the enablers and barriers of the cascade-of-care. We linked these barriers/facilitators to each step of the cascade as they were reported by the original manuscripts. If a manuscript reported several steps of the cascade and did not specify in which step the facilitator/barrier were important, we classified as multistage.

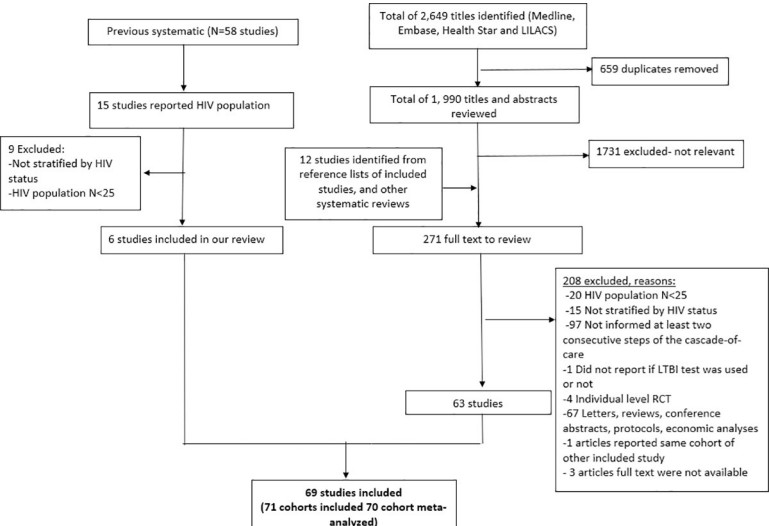

**Fig 3. PRISMA flow diagram.** LTBI, latent tuberculosis infection; RCT, randomized clinical trial.

## Results

As shown in Fig 3, 2,649 titles were identified in our updated search. Among them, 271 full texts were screened for eligibility, and we included 51 studies. In addition, we identified 6 studies from the previous cascade-of-care systematic review, and 12 more studies were identified from other systematic reviews or reference lists of included studies. No manuscript was excluded on the basis of language criteria. In total, 69 studies [17–85] were included. Of the 69 studies, 2 reported more than one cohort [17,76], yielding 71 cohorts. Among those, 70 cohorts were meta-analyzed, comprising 94,011 PLHIV. One manuscript [85] not included in the meta-analyses reported national data from 16 low- and middle-income countries, supported by the US President's Emergency Plan for AIDS Relief (PEPFAR). Due to the particularity of financial support (which might not reflect the reality of the other cohorts), the large study populations (over 1.8 million PLHIV starting TPT), and limited other information to characterize these cohorts, we summarized the PEPFAR outcomes separately.

Table 1 summarizes the main characteristics of the cohorts included in the meta-analyses. Sixty-eight (97%) cohorts were observational studies, and the 2 remaining cohorts were derived from an RCT [76]. Sixty-two cohorts reported the type of clinic where PLHIV were

**Table 1. Summary of cohorts included in the meta-analyses (N = 70)[1].**

| Factor/Parameter | Cohorts, (N, %) | Participants (N, %)[2] |
|---|---|---|
| Overall | 70 (100.0%) | 94,011 (100.0%) |
| Population | | |
| Children | 3 (4.3%) | 53,538 (56.9%) |
| Adults | 35 (50.0%) | 24,431 (26.0%) |
| Both | 20 (28.6%) | 1,048 (1.1%) |
| Unclear | 12 (17.1%) | 14,994 (15.9%) |
| Study design | | |
| Cluster RCT[3] | 2 (2.9%) | 3,024 (3.2%) |

(*Continued*)

**Table 1.** (Continued)

| Factor/Parameter | Cohorts, (N, %) | Participants (N, %)[2] |
|---|---|---|
| Cross-sectional | 7 (10.0%) | 17,955 (19.1%) |
| Pre-post study | 1 (1.4%) | 1,395 (1.5%) |
| Prospective cohort | 38 (54.3%) | 38,518 (41.0%) |
| Retrospective cohort | 22 (31.4%) | 33,119 (35.2%) |
| Country by World Bank definition[4] | | |
| High-income | 25 (35.7%) | 46,340 (49.3%) |
| Low- and middle-income | 44 (62.9%) | 46,903 (49.9%) |
| WHO regions | (100.0%) | |
| Africa | 21 (30.0%) | 27,460 (29.2%) |
| America | 18 (25.7%) | 33,139 (35.3%) |
| Europe | 13 (18.6%) | 21,198 (22.5%) |
| Southeast Asia | 4 (5.7%) | 1,905 (2.0%) |
| Western Pacific | 14 (20.0%) | 10,309 (11.0%) |
| Type of care | | |
| HIV clinic | 40 (57.2%) | 67,176 (71.5%) |
| TB clinic | 4 (5.7%) | 408 (0.4%) |
| Mixed HIV/TB care (majority primary clinics offering TB/HIV services) | 10 (14.3%) | 12,645 (13.5%) |
| Other (specific population, e.g., prisons, PWID users) | 8 (11.4%) | 8,558 (9.1%) |
| Unclear | 8 (11.4%) | 5,224 (5.6%) |
| Used LTBI tests (N = 49) | | |
| IGRA or TST | 15 (30.6%) | 24,000 (36.2%) |
| Only IGRA | 12 (24.5%) | 6,497 (9.8%) |
| Only TST | 22 (44.9%) | 35,722 (53.9%) |
| Did not use LTBI tests (N = 21) | | |
| Only symptoms screen | 16 (76.2%) | 22,757 (81.9%) |
| Symptoms screened AND chest X-ray (or other diagnostic tests)[5] for eligibility of TPT | 2 (9.5%) | 1,813 (6.5%) |
| Not clear if used additional tests | 3 (14.3%) | 3,222 (11.6%) |
| LTBI regimen used | | |
| Isoniazid regimen | 50 (71.4%) | 76,547 (81.4%) |
| 3 months of rifampin and isoniazid[6] | 1 (1.5%) | 304 (0.3%) |
| Mainly isoniazid regimen but fewer patients used other regimens (RBT-PZA, Rif-PZA, RIF-INH-PZA)[7] | 5 (7.1%) | 8,973 (9.5%) |
| Not specified | 14 (20.0%) | 8,187 (8.7%) |

[1]PEPFAR report [85] not included in this table.

[2]Denominator is the overall population; N = 94,011.

[3]One RCT stratified in 2 cohorts.

[4]One multicenter study, in different countries, with different income classification, not included here.

[5]One study used Xpert regardless the presence of symptoms, and other study used chest X-ray regardless the symptoms.

[6]92% cohort used 3 months of isoniazid and rifampin.

[7]More than 80% of patients of these cohorts used isoniazid regimen.

IGRA, interferon release gamma assay; INH, isoniazid; LTBI, latent tuberculosis infection; N, Number; PEPFAR, President's Emergency Plan for AIDS Relief; PWID, persons who inject drugs; PZA, pyrazinamide; RBT, rifabutin; RCT, randomized clinical trial; RIF, rifampin; TB, tuberculosis; TPT, tuberculosis preventive therapy; TST, tuberculin skin test; WHO, World Health Organization.

evaluated for TPT, and 40 (64%) of these cohorts were seen in HIV clinics. Twenty-one cohorts [17,66–79] did not use LTBI tests, while 49 cohorts [17–55] used LTBI tests; 22 used only TST, 12 used only IGRA, and 15 used either IGRA or TST. Among the 56 (80%) studies that reported the type of TPT regimen, mono-isoniazid regimens were the primary regimen prescribed in 50 cohorts (89%), and only one (2%) of the included cohorts primarily prescribed rifamycin-based short regimens (3 to 4 months of rifampicin and isoniazid) [29]. Additional details on the included studies are reported in S3–S5 Tables.

Demographic and clinical information by cohort is shown in S5 Table. Age was reported in 37 (52%) cohorts, and the median age was 38.0 [interquartile range (IQR), 34.0;43.6]. Sex was reported in 59 (84%) cohorts, and 65.9% (46,328 /70,297) were men. CD4 cell count was reported in 31 (44%) cohorts, and the median count was 390 cell/mm3 (IQR 312;458). Fifty-three (76%) cohorts reported the use of ART with 43.6% (29,629/67,947) of PLHIV being treated with ART.

S1A Fig shows the quality assessment, and S1B Fig lists the evaluation of the 68 observational studies. For population selection, 67% (46/68) of studies were classified as high risk of bias, most (41/68; 60%) due to the use of convenience sampling or because the sampling method was not described. For outcome ascertainment, 16% (11/68) of studies were classified as either high or unclear risk of bias. For exposure ascertainment, 13% (9/68) of studies did not report this information. Only one study was a cluster RCT [76], and in all domains evaluated, it was classified as low risk of bias.

In the cascade-of-care analyses, all the cohorts that did not use LTBI tests prior to treatment initiation were in low- and middle-income countries (Tables 2 and S6). Out of all PLHIV identified, the cumulative proportion starting and completing TPT was 40.9% (95% CI: 39.3% to 42.7%) and 33.2% (95% CI: 31.6% to 34.9%), respectively. The main losses occurred at the step of provider recommendation of TPT (pooled estimate of 66.2%, representing a loss of 33.8%). Among cohorts that used LTBI tests (Tables 2 and S7), the cumulative proportion of PLHIV starting and completing TPT was 60.4% (95% CI: 58.1% to 62.6%) and 41.9% (95% CI:39.6% to 44.2%), respectively. For these cohorts, the main losses were in the provider recommendation of TPT and completion of TPT. Using random effect model, the main losses (S8 Table) remained in the same steps; however, the cumulative proportion of patient completing TPT was 54.0% (95% CI: 12.6% to 76.9%) among cohorts that did not receive LTBI tests and 60.3% (95% CI:37.7% to 75.0%) among cohorts that received LTBI tests.

To explore possible reporting bias among studies reporting only a limited number of steps of the cascade-of-care, we analyzed the cohorts that reported data for all steps. Similar results were found with a cumulative TPT completion rate of 32.4% (95% CI: 30.0% to 34.9%) among cohorts that did not use LTBI tests to determine TPT eligibility and 35.0% (95% CI: 32.8% to 40.6%) among cohorts that used these tests (Table 2). Using random effect model among cohorts that reported all steps, the cumulative proportion of patients completing TPT was 42.6% (95% CI: 2.6% to 74.5%) among cohorts that did not receive LTBI tests and 53.1% (95% CI: 18.4% to 79.9%) among cohorts that received LTBI tests (S8 Table).

To explore the variability in the estimates of losses at different steps, we generated forest plots (S2 and S3 Figs). The variability between studies was high in all proportions presented, regardless of the use of LTBI tests.

Among the cohorts that used LTBI tests, 25 cohorts were from high-income countries, 23 were from low- and middle-income countries, and one multicenter cohort included sites from both settings [40]. The pooled prevalence of LTBI was 13.2% in cohorts from high-income countries, and 26.2% within low- and middle-income countries. When comparing the 2 settings, the losses occurred in different steps over the cascade-of care, but, consistently, the step with the greatest losses was TPT completion (Table 3). The cumulative proportion of patients

**Table 2. Pooled estimate for each step of the cascade-of-care[1] (pooled using fixed effect model).**

| Steps | All cohorts | | | | Cohorts that reported all cascade-of-care steps | |
|---|---|---|---|---|---|---|
| | Cohorts | n/N | Pooled estimate (95 CI%) | Pooled estimate (95 CI%) of cumulative percentage retained in the cascade | Cohorts | Calculated cumulative percentage retained in the cascade (95 CI%)[2] |
| *Did not use LTBI tests (n = 21 cohorts)[3]* | | | | | | |
| Proportion 1: Had a medical evaluation/Identified | 15 | 10,806/ 13,552 | 79.7% (95% CI: 79.1% to 80.4%) | 79.7% (95% CI: 79.1% to 80.4%) | 6 | 76.1% (95% CI: 75.0% to 77.2%) |
| Proportion 2: Recommended TPT/ med evaluation | 10 | 5,464/ 7,040 | 77.6% (95% CI: 76.6% to 78.6%) | 61.8% (95% CI: 60.6% to 63.2%) | 6 | 59.1% (95% CI: 57.3% to 60.9%) |
| Proportion 3: Started TPT/ Recommended TPT | 10 | 3,259/ 4,922 | 66.2% (95% CI: 64.9% to 67.5%) | 40.9% (95% CI: 39.3% to 42.7%) | 6 | 44.3% (95% CI: 42.1% to 46.6%) |
| Proportion 4: Completed TPT treatment/Started TPT | 13 | 8,064/ 9,937 | 81.2% (95% CI: 80.4% to 81.9%) | 33.2% (95% CI: 31.6% to 34.9%) | 6 | 32.4% (95% CI: 30.0% to 34.9%) |
| *Used LTBI tests (n = 49 cohorts)* | | | | | | |
| Proportion 1: Initiated LTBI testing/ Identified | 34 | 30,813/ 35,201 | 87.5% (95% CI: 87.2% to 87.9%) | 87.5% (95% CI: 87.2% to 87.9%) | 8 | 68.8% (95% CI: 67.7% to 70.0%) |
| Proportion 2: Completed LTBI testing/initiated LTBI test | 37 | 32,899/ 34,447 | 95.5% (95% CI: 95.3% to 95.7%) | 83.6% (95% CI: 83.1% to 84.1%) | 8 | 63.6% (95% CI: 62.0% to 65.2%) |
| *Prevalence of LTBI positive*: Tests positive/completed test | 47[4] | 10,131/ 55,587 | 18.2% (95% CI: 17.9% to 18.5%) | - | | - |
| Proportion 3: Medical evaluation completed/Needed medical evaluation | 31 | 2,839/ 2,881 | 98.5% (95% CI: 98.0% to 98.9%) | 82.3% (95% CI: 81.4% to 83.2%) | 8 | 63.6% (95% CI: 61.0% to 65.1%)[5] |
| Proportion 4: Recommended TPT/ Medical evaluation completed | 24 | 2,308/ 2,714 | 85.0% (95% CI: 83.6% to 86.3%) | 70.0% (95% CI: 68.1% to 71.8%) | 8 | 54.3% (95% CI: 50.8% to 56.8%) |
| Proportion 5: Started TPT/ Recommended LTBI treatment | 23 | 4,583/ 5,313 | 86.3% (95% CI: 85.3% to 87.2%) | 60.4% (95% CI: 58.1% to 62.6%) | 8 | 49.5% (95% CI: 48.2% to 55.2%) |
| Proportion 6: Completed TPT/ Started TPT treatment | 23 | 3,762/ 5,419 | 69.4% (95% CI: 68.2% to 70.6%) | 41.9% (95% CI:39.6% to 44.2%) | 8 | 35.0% (95% CI: 32.8% to 40.6%) |

[1]Pooled using fixed effect model.

[2]This value is the product of the cumulative percentage from the preceding step, multiplied by the pooled estimate from this step.

[3]All cohorts from low- to middle-income countries.

[4]Among 49 cohorts, 47 reported the positivity rates of LTBI tests.

[5]Confidence intervals estimated using inverse method.

CI, confidence interval; LTBI, latent tuberculosis infection; N, Number; TPT, tuberculosis preventive therapy.

completing TPT were similar in high- and low- and middle-income countries, 37.9% (95% CI: 34.1% to 41.2%) and 42.9% (95% CI: 39.8% to 45.9%), respectively (Table 3). Using random effect model (S9 Table), the cumulative proportion of patients completing TPT in high- and low- and middle-income countries were 43.7% (95% CI: 17.6% to 66.0%) and 72.4% (95% CI: 9.2% to 89.1%), respectively.

In the stratified analysis by type of LTBI test performed, the losses in the different steps of the cascade were variable between the cohorts that used TST or IGRA. But, at the end of the cascade-of-care, the cumulative proportion of patients completing TPT was similar as seen in S4 Fig.

Among cohorts that did not receive LTBI tests, the cumulative TPT initiation and completion were similar if PLHIV were followed at HIV clinics or other clinics (S5 Fig). However, among the cohorts that received LTBI tests (S6 Fig), TPT completion was higher among PLHIV that were followed in HIV clinics, compared to other clinics [54.4% (95% CI: 29.1% to 71.5%) versus 52.3% (95% CI 1.3% to 82.9%)].

**Table 3. Pooled estimate for each step of the cascade in cohorts that used LTBI tests, stratified by country income level[1] (pooled using fixed effect model).**

| Steps | Cohorts | n/N | Pooled estimate | Calculated cumulative percentage retained in the cascade (95 CI%)[2] |
|---|---|---|---|---|
| *High-income countries (N = 25 cohorts)* | | | | |
| Proportion 1: Initiated LTBI testing/Identified | 19 | 23,239/ 26,288 | 88.4% (95% CI: 88.0% to 88.8%) | 88.4% (95% CI: 88.0% to 88.8%) |
| Proportion 2: Completed LTBI testing/initiated LTBI test | 19 | 22,778/ 23,239 | 98.0% (95% CI: 97.8% to 98.2%) | 86.9% (95% CI: 86.1%. to 87.2%) |
| *Prevalence of LTBI test positive: Positive/completed test* | 24 | 4,551/ 34,506 | 13.2% (95% CI: 12.8% to 13.6%) | - |
| Proportion 3: Medical evaluation completed/Needed medical evaluation | 17 | 750/763 | 98.3% (95% CI: 97.1% to 99.0%) | 85.4% (95% CI: 83.6% to 86.2%) |
| Proportion 4: Recommended TPT/Medical evaluation completed | 11 | 570/641 | 88.9% (95% CI: 86.3% to 91.1%) | 75.9% (95% CI: 72.1% to 78.6%) |
| Proportion 5: Started TPT/Recommended LTBI treatment | 11 | 1,486/1,666 | 89.2% (95% CI: 87.6% to 90.6%) | 67.7% (95% CI: 63.2% to 71.3%) |
| Proportion 6: Completed TPT/Started TPT | 12 | 1,375/2,461 | 55.9% (95% CI: 53.9% to 57.8%) | 37.9% (95% CI: 34.1% to 41.2%) |
| *Low- and middle-income countries (N = 23 cohorts)* | | | | |
| Proportion 1: Initiated LTBI testing/Identified | 14 | 6,806/8,145 | 83.6% (95% CI: 82.7% to 84.3%) | 83.6% (95% CI: 82.7% to 84.3%) |
| Proportion 2: Completed LTBI testing/initiated LTBI test | 17 | 9,463/ 10,440 | 90.6% (95% CI: 90.1% to 91.2%) | 75.7% (95% CI: 74.5% to 76.9%) |
| *Prevalence of LTBI test positive: Positive/completed test* | 22 | 5,341/ 20,423 | 26.2% (95% CI: 25.6% to 26.8%) | - |
| Proportion 3: Medical evaluation completed/Needed medical evaluation | 14 | 2,089/2,118 | 98.6% (95% CI: 98.0% to 99.0%) | 74.7% (95% CI: 73.0% to 76.1%) |
| Proportion 4: Recommended TPT/Medical evaluation completed | 13 | 1,738/2,073 | 83.8% (95% CI: 82.2% to 85.4%) | 62.6% (95% CI: 60.0% to 65.0%) |
| Proportion 5: Started TPT/Recommended LTBI treatment | 12 | 3,097/3,647 | 84.9% (95% CI: 83.7% to 86.0%) | 53.1% (95% CI: 50.2% to 55.9%) |
| Proportion 6: Completed TPT/Started TPT | 11 | 2,387/2,958 | 80.7% (95% CI: 79.2% to 82.1%) | 42.9% (95% CI: 39.8% to 45.9%) |

[1]Pooled using fixed effect model. One multicenter study, in different countries with different income classification, not included in these analyses since the cascade steps were not reported by center (Sester and colleagues [40]).

[2]This value is the product of the cumulative percentage from the preceding step, multiplied by the pooled estimate from this step.

CI, confidence interval; LTBI, latent tuberculosis infection; N, Number; TPT, tuberculosis preventive therapy.

S10 Table summarizes the results of the PEPFAR program results during the years of 2017 to 2019, in 14 African countries, plus Haiti and Vietnam. All PLHIV that started TPT were receiving ART. A total of 1,805,145 PLHIV started TPT, of whom 59.8% completed it. Kenya, Nigeria, South Africa, and Tanzania were the countries with higher number of PLHIV starting and completing TPT.

As shown in S11 Table, the overall pooled proportion of PLHIV eligible for TPT who completed treatment was 26.5% (95% CI: 18.9% to 35.9%). Use of LTBI tests, country-level income, and type of service were not significantly associated with this outcome, in meta-regression.

Table 4 summarizes enablers reported in 17 studies [18,21,24,38,51,58,66,70–76,78,79,82]. The most common facilitators were related to the initial steps (identification, initial LTBI testing, and completing LTBI testing) and to initiation and completion of TPT. Regardless of the steps, most facilitators were from the health system perspective and included activities such as training healthcare workers about the importance of TPT in PLHIV and proper techniques for injection and reading of TST. Integration of HIV and TPT care was a facilitator for multiple

**Table 4. Enablers and barriers for different steps of the cascade-of-care identified in the studies included in the review.**

| | Enablers (N = 17 cohorts) | Barriers (N = 12 cohorts) |
|---|---|---|
| Multistage | • HIV and TPT care integrated [18,38,72,74,76,79,82] | • Fragmentation of care of HIV and TB patients [34,81]<br>• Testing and TPT were implemented by TB programs that were not familiar with the care of HIV patients [66]<br>• Stigma of HIV patients receiving care in TB clinics [72,74]<br>• Lack of information, motivation, and support to HCW [21,80,81]<br>• Too much workload [81] |
| Identification | • Participation of staff in the design of TPT implementations strategy [70]<br>• Patient education material [70]<br>• HIV testing for household contacts of infectious TB disease [38,71]<br>• Community HCWs initiating contact [71]<br>• TB/HIV care in antenatal care services [73,75]<br>• Creation of a TB/HIV integration officer and a TB screening officer [76]<br>• Task shifting TB screening to primary care [76] | • None Identified |
| Initial testing | • Theoretical training on TST [18,24]<br>• TST training [18,24,70]<br>• Extra consultation room [18]<br>• TB screening by "lay counselor" [18]<br>• TB screening done by physicians [24]<br>• TB nurse dedicated for administration of TST [24]<br>• TB counseling at the moment of HIV diagnosis [21]<br>• Socials workers who traced and visited patients who missed appointments [21] | • Gaps in infection control: absence of N95 masks and training for HCW not conducted regularly [72]<br>• Establishing a cold chain for TST [24]<br>• HCW confusion between TST and BCG vaccine [24]<br>• HCW has no time to perform TST [70]<br>• HIV test councilors who had limited familiarity with LTBI [21] |
| Received test results | • Calling patients to return for TST reading [18]<br>• HCW explained when and why patients need to return [58]<br>• Skin tested interpreted in closer facility or at patient's home or work [58] | • Difficult to motivate patients to return for TST reading [70]<br>• TST that could not be interpreted [34]<br>• Long wait time for patients or inconvenient clinic hours [58]<br>• Travel costs to return to clinics [58]<br>• Absence of work or family duties [58]<br>• Patient not told or did not understand that had to return [58] |
| Needed medical evaluation | • None identified | • None identified |
| Medical evaluation | • None identified | • None identified |
| Recommended TPT treatment | • None identified | • None identified |
| Started TPT | • Staff training about TPT [51,66]<br>• Patients were informed about adverse event of IPT and disadvantages of stopping treatment [66]<br>• TPT paper tool to facilitate HCWs in charge to prescribe TPT [74]<br>• Clinic received list of patients eligible for TPT [51] | • Counseling patients about TPT is difficult and time consuming [70]<br>• Patients not knowledgeable about TPT [70]<br>• Poor adherence to ART and follow-up [74]<br>• Pill burden [34,74]<br>• Lack of liver function tests at baseline [74]<br>• Fear of side effects [74] |
| Completed TPT | • Supervision of children who required TPT [79]<br>• Patients could choose model for TPT delivery coordinated with ART refills [78,82] | • Adverse drug reactions—hepatotoxicity in the context of hepatitis B infection and alcohol use [18]<br>• Neuropathy developing in the context of INH use and vitamin B6 not being prescribed [18]<br>• Supply chain and drug stockouts [72,78]<br>• Follow-up of patients on TPT is difficult and time consuming [70] |

BCG, Bacillus Calmette–Guerin; HCW, health care workers; INH, isoniazid; LFT, liver functions tests; TB, tuberculosis; TPT, tuberculosis preventive therapy; TST, tuberculosis skin test.

steps in 7 studies [18,38,72,74,76,79,82]. Barriers were reported by 12 studies [18,21,24,34,58, 66,70,72,74,78,80,81], primarily at the initial steps of identification and testing and the final steps of initiating and completing TPT. Nonintegrated TB and HIV care was a barrier for at multiple steps [34,81]. Pill burden, fear of adverse events, and stocks out of LTBI drugs were also reported as barriers for starting and completing TPT [18,34,72,74,78].

## Discussion

In this meta-analysis exclusively in PLHIV, we found that cumulative TPT completion was similar in studies that used or did not use LTBI tests and also similar in studies from high- or low- and middle-income income settings, regardless of use of LTBI tests. Despite the losses in multiples stages, overall TPT completion was better than overall completion observed in an earlier review that included multiple at-risk populations [11]. Health system facilitators included training of healthcare workers for TPT, and integration of TB and HIV care, while barriers included fear of adverse events, pill burden, and lack of knowledge among healthcare workers and patients.

Our study has several public health implications. Despite major losses in the cascade-of-care found in our analyses, the overall initiation and completion of TPT among PLHIV was higher than described in a previous systematic review [11]—which evaluated multiple at-risk populations. The differences in the study populations included in these 2 systematic reviews might explain the difference in findings. Our systematic review was exclusively in PLHIV, who are usually already linked to the healthcare system. In the previous systematic review, the main loss (approximately 28%) occurred in the initial identification and linkage to healthcare step [11], likely because the populations in that review were mainly contacts and immigrants, who were not already linked to the healthcare system.

The important losses in the cascade found might be explained by the fragmentation of TB and HIV care. Multiple studies reported that integration of TB and HIV care was an important enabler of TPT [18,38,72,74,76,79,82], while fragmentation of TB and HIV care was identified as an important barrier [34,81]. This suggests that policymakers should work to close the gap between HIV and TB care.

An important finding in our review was that the use of LTBI tests was not a barrier to TPT initiation or completion among cohorts that used them. Cohorts that used IGRA, compared to cohorts tested with TST, had a higher percentage of population in initiating (87% versus 73%) and completing (99% and 89%) these tests, but overall TPT initiation and completion was similar, regardless of type of test used.

Despite the possible lower sensitivity of IGRAs and TST in PLHIV, previous trials and systematic reviews [2,3,5] have shown that PLHIV with positive LTBI tests benefitted most from TPT, since the risk of TB in PLHIV with a positive LTBI test (TST or IGRA) is 11-fold higher than in PLHIV with a negative LTBI test [5]. Interestingly, TPT recommendation and initiation was higher among cohorts that used LTBI tests; this may reflect providers and patients' beliefs in prescribing and accepting treatment with evidence of a positive test. For these reasons, we suggest that the use of LTBI tests should be encouraged, not only in high-income countries where it is already part of care, but also in low- and middle-income countries, where this review found numerous reports of its successful use. TPT may provide some benefit in high TB incidence settings if all PLHIV are treated without use of LTBI testing [86,87]. However, the use of LTBI tests can identify those most likely to benefit [5], and treatment without prior LTBI testing might expose PLHIV without TB infection to a nontrivial risk of adverse events [88,89]. Furthermore, the healthcare expenditures for drugs, follow-up visits, and tests including those related to AE, to provide TPT to PLHIV who may not benefit from this could be redirected to strengthening the LTBI cascade-of-care in those (with positive LTBI tests) who will benefit more from TPT.

Completing the medical evaluation was considered an important barrier in cohorts that did not use LTBI testing. These cohorts used diagnostic algorithm strategies [4] that rely on symptom screening. However, in the presence of symptoms and/or if the patient is receiving ART, a chest X-ray is recommended before TPT initiation [4]. All these cohorts were from low- and

middle-income countries, where chest X-ray services are not commonly accessible. Even where the test is available, the cost often falls on the patient and their family [90] and can be prohibitively expensive. Therefore, the elimination of the financial burden of chest X-rays is essential for TPT scale-up or alternative algorithms using other diagnostic tests to exclude active TB [91,92].

Finally, among the 50 cohorts that provided information on the TPT regimen prescribed, 49 used isoniazid, even though short rifamycin regimens have been available for over 2 decades. TPT completion was low in primary and all stratified analyses, and pill burden and fear of adverse event were reported as barriers for TPT initiation and completion. Certain ART regimens may present drug–drug interactions with rifamycin regimen, especially protease inhibitors. This could explain the lower prescription of rifamycin short regimens by the providers. However, non-nucleoside reverse transcriptase inhibitors (such as efavirenz) and the integrase inhibitors—such as raltegravir or dolutegravir (doubling the dose)—can be coadministered with rifamycins [14]. Increase use of shorter rifamycin-based regimens should be considered, as these may improve TPT completion and are safer, cheaper, and at least as effective as isoniazid regimens [93–99].

This systematic review has a number of limitations. Only 14 of the included cohorts reported all the steps of the cascade-of-care. To include a greater diversity of study settings, we included all 71 cohorts in which at least 2 consecutive cascade steps were reported. This allowed us to calculate the proportion of PLHIV retained in multiple steps of the cascade-of-care from a much larger number of studies, enhancing generalizability. When we compared results of analysis of all cohorts with the 14 studies that included all the steps of the cascade-of-care, results were very similar. Barriers and facilitators were not systematically reported in all included cohorts, so we could not fully understand why the losses and/or retention occurred at each cascade step. All but one of the studies were observational and mostly used convenience sampling or did not describe the population selection. Hence, the majority of studies were judged to have potential selection bias. As a result, we consider the overall quality of evidence to be low, limiting inferences from our findings. Only 3 studies focused exclusively on children, so the pediatric TPT cascade-of-care could not be assessed.

The strengths of this review include the large number of cohorts meta-analyzed ($N = 70$) and the large population of PLHIV ($N = 94,011$), which allowed us to perform more detailed stratified analyses including country income level, use of LTBI tests, type of LTBI test, and type of clinic. We also evaluated cohorts from different countries, with a wide range of socioeconomic status and resource availability, enhancing the generalizability of our findings.

## Conclusions

In conclusion, TPT initiation and completion were higher in PLHIV than previously reported for other at-risk populations. Linkage to the health system, clear and consistent evidence from multiple randomized trials of the benefits of TPT, and consistent recommendations by international and national public health authorities might explain this degree of relative success. These lessons should be applied in other groups, particularly in household contacts. Despite this, our analysis of the LTBI cascade-of-care among PLHIV reveals continued important losses. Only 40% of PLHIV eligible for TPT completed this, which is much lower than other care targets in HIV, such as the famous "90-90-90" [100]. Therefore, continued efforts are needed to further improve the LTBI cascade-of-care in this population.

## Supporting information

**S1 PRISMA Checklist. PRISMA checklist for reporting systematic reviews and meta-analyses.** (DOCX)

**S1 Search Strategy. Search Strategy (Medline Ovid and LILACS).**
(DOCX)

**S1 Table. Quality assessment tool used in review for observational studies (adapted from Newcastle–Ottawa Scale).**
(DOCX)

**S2 Table. Quality assessment tool used in review for cluster randomized trials (adapted from Cochrane RoB tool).**
(DOCX)

**S3 Table. Summary of design features of the studies included in the review.**
(DOCX)

**S4 Table. Characteristics of participants in the studies included in the review.**
(DOCX)

**S5 Table. Number and clinical characteristics of participants in the studies included in the review.**
(DOCX)

**S6 Table. Number of participants in each step of the cascade-of-care among studies that did not use LTBI tests.**
(DOCX)

**S7 Table. Number of participants in each step of the cascade-of-care among studies that used LTBI tests.**
(DOCX)

**S8 Table. Sensitivity analysis.** Pooled estimate for each step of the cascade-of-care, using random effect model.
(DOCX)

**S9 Table. Sensitivity analysis table.** Pooled estimate for each step of the cascade in cohorts that used LTBI tests, stratified by country income level[1] random effect model.
(DOCX)

**S10 Table. Summary results of the report US President's Emergency Plan for AIDS Relief, 2017–2019 (PEPFAR) Tuberculosis Preventive Treatment Scale-Up Among Antiretroviral Therapy Patients.**
(DOCX)

**S11 Table. Meta-regression of patients completing TPT over PLHIV identified.**
(DOCX)

**S1 Fig. Quality assessment of the studies included in the review.**
(DOCX)

**S2 Fig. Forest plots among studies that did not use LTBI tests.** LTBI, latent tuberculosis infection; TPT, tuberculosis preventive therapy.
(DOCX)

**S3 Fig. Forest plots among studies that used LTBI tests.** LTBI, latent tuberculosis infection; TPT, tuberculosis preventive therapy.
(DOCX)

**S4 Fig. Cumulative proportion of each step of the cascade among cohorts that did not use LTBI test stratified by type of clinic where PLHIV were evaluate.** Pooled using fixed effect model. IGRA, interferon gamma release assay; LTBI, latent tuberculosis infection; PLHIV, people living with HIV; TPT, tuberculosis preventive therapy; TST, tuberculin skin test.
(DOCX)

**S5 Fig. Cumulative proportion of each step of the cascade among cohorts that used LTBI test stratified by type of clinic where PLHIV were evaluated.** Pooled using fixed effect model. LTBI, latent tuberculosis infection; PLHIV, people living with HIV.
(DOCX)

**S6 Fig. Cumulative proportion of each step of the cascade among cohorts that used LTBI test stratified by type of clinic where PLHIV were evaluated ($N$ = 49 cohorts).** Pooled using fixed effect model. LTBI, latent tuberculosis infection; PLHIV, people living with HIV; TPT, tuberculosis preventive therapy.
(DOCX)

**S1 Data. Data used to perform the meta-analyses.**
(XLSX)

## Author Contributions

**Conceptualization:** Mayara Lisboa Bastos, Jonathon R. Campbell, Olivia Oxlade, Dick Menzies.

**Data curation:** Mayara Lisboa Bastos, Luca Melnychuk, Dick Menzies.

**Formal analysis:** Mayara Lisboa Bastos, Luca Melnychuk, Jonathon R. Campbell, Olivia Oxlade, Dick Menzies.

**Funding acquisition:** Olivia Oxlade, Dick Menzies.

**Investigation:** Mayara Lisboa Bastos, Luca Melnychuk, Jonathon R. Campbell, Olivia Oxlade, Dick Menzies.

**Methodology:** Mayara Lisboa Bastos, Luca Melnychuk, Jonathon R. Campbell, Olivia Oxlade, Dick Menzies.

**Project administration:** Olivia Oxlade.

**Resources:** Olivia Oxlade, Dick Menzies.

**Software:** Mayara Lisboa Bastos, Jonathon R. Campbell.

**Supervision:** Dick Menzies.

**Validation:** Mayara Lisboa Bastos, Luca Melnychuk, Jonathon R. Campbell, Olivia Oxlade, Dick Menzies.

**Visualization:** Mayara Lisboa Bastos, Luca Melnychuk, Jonathon R. Campbell, Olivia Oxlade, Dick Menzies.

**Writing – original draft:** Mayara Lisboa Bastos, Dick Menzies.

**Writing – review & editing:** Mayara Lisboa Bastos, Luca Melnychuk, Jonathon R. Campbell, Olivia Oxlade, Dick Menzies.

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
