## [Editor Report · Decision Letter 0]

6 Jan 2021

Dear Dr Menzies, 

Thank you for submitting your manuscript entitled "The Latent Tuberculosis Cascade-of-Care Among People Living with HIV: A Systematic Review and Meta-Analysis" for consideration by PLOS Medicine.

Your manuscript has now been evaluated by the PLOS Medicine editorial staff and I am writing to let you know that we would like to send your submission out for external peer review.

Kind regards,

Thomas J McBride, PhD

Senior Editor

PLOS Medicine

---

## [Decision Letter · Decision Letter 1]

17 Feb 2021

Dear Dr. Menzies,

Thank you very much for submitting your manuscript "The Latent Tuberculosis Cascade-of-Care Among People Living with HIV: A Systematic Review and Meta-Analysis" (PMEDICINE-D-20-06012R1) for consideration at PLOS Medicine. 

Your paper was evaluated by the editors here and sent to independent reviewers, including a statistical reviewer. The reviews are appended at the bottom of this email and any accompanying reviewer attachments can be seen via the link below:

[LINK]

In light of these reviews, we will not be able to accept the manuscript for publication in the journal in its current form, but we would like to invite you to submit a revised version that addresses the reviewers' and editors' comments fully. You will appreciate that we cannot make a decision about publication until we have seen the revised manuscript and your response, and we expect to seek re-review by one or more of the reviewers. 

We hope to receive your revised manuscript by Mar 10 2021 11:59PM. Please email us (plosmedicine@plos.org) if you have any questions or concerns.

Please let me know if you have any questions, and otherwise we look forward to receiving your revised manuscript. 

Sincerely,

Richard Turner, PhD

rturner@plos.org

Please resubmit your article as a "Research article" (which is standard for systematic reviews and meta-analyses at PLOS Medicine), and we will assign an academic editor to the paper upon resubmission. 

Please update the search, e.g., to the start of 2021. 

Please combine the "Methods" and "Findings" subsections of your abstract. The final sentence of the new combined subsection should begin "Study limitations include ..." or similar, and should quote 2-3 of the study's main limitations. 

Please quote some information on the cohorts in the abstract, e.g., the number or proportion in high-income countries and so on; and on the designs of the included studies. Are you able to quote aggregate demographic details for participants?

After the abstract, please add a new and accessible "author summary" section in non-identical prose. You may find it helpful to consult one or two recent research articles published in PLOS Medicine to get a sense of the preferred style. 

Please revisit "was better" at line 222, explaining in a sentence, say, the respects in which coverage was "better" and perhaps quoting quantitative findings. We also suggest reiterating the patient group in this sentence.

In your Discussion section, please reverse the order of the "strengths" and "limitations" paragraphs. 

Throughout the text, please style reference call-outs as follows: " ... one cohort [17,66]." (i.e., no spaces within the square brackets). Please ensure that call-outs fall before punctuation, e.g., at line 223.

In the reference list, please abbreviate journal names consistently (e.g., "Lancet" will suffice for reference 2). Please remove extraneous information on competing interests, e.g., from reference 9. 

Please remove the attached Collection proposal form.

Please supply a completed PRISMA checklist with your revision, labelled "S1_PRISMA_Checklist" or similar and referred to in the Methods section of your main text. In the checklist, please refer to individual items by section (e.g., "Methods") and paragraph number rather than by page or line numbers, as the latter generally change in the event of publication. 

Comments from the reviewers:

*** Reviewer #1: 

[See attachment]

Michael Dewey

*** Reviewer #2: 

PMEDICINE-D-20-06012R1 

Overall summary of critique

Bastos et al. present a well conducted systematic review and meta-analysis evaluating the cascade of latent TB (LTBI) care in PLHIV. This is provides important data to quantify the gaps in care with additional nuance regarding the outcomes with and without the use of LTBI tests. My main points of critique are that the authors should provide more information regarding how they collected data to include in the narrative review (as this could be a separate topic) and about the framing of the conclusion as emphasizing that the outcomes for PLHIV were better than for other high-risk populations because these outcomes remain below target and this risks an implicit judgement that detracts from the need to emphasize the need to close these gaps in this high risk population as well as others. The discussion should also include some of the nuances regarding the imperfect sensitivity of current LTBI testing in PLHIV and limited knowledge re: ART drug interactions with newer shorter regimens.

Reviewer comments by section

Abstract

Line 55-56 - I think this second conclusion sentence should be split in two as think you are saying the use of LTBI testing did not reduce the proportion of eligible PLHIV who completed TPT?

Introduction

Line 77 - please clarify 'never received it' - does this mean did not receive a course of TPT or did not actually initiate TPT?

Methods

Lines 147-148: narrative synthesis methodology requires further information. Currently this appears to be a throwaway comment but it is important to understand how the authors approached data collection to answer this question.

Results

Line 160 - were TST thresholds consistent i.e. always <5mm and were IGRA results always classified as positive only according to the manufacturer's recommendations?

Line 180 - re: main loss being provider recommendation for TPT- did studies always actually measure this/refer to this in the same way? Where does the 31.1% come from (don't see it in Table 2)?

Lines 197-199 - please review phrasing here re: clarity around despite where the losses occurred?

Discussion

Line 222- would include HIV in first sentence and currently it is a bit vague - do you mean better than the overall rate observed as per the Alsdurf review or for the subgroup with HIV? I think this refers to the former but see comment re: overall conclusion. I think framing should be although considerably better than for other high risk groups, analysis of the LTBI cascade of care for PLHIV reveals large gaps at multiple stages. This is important because there is generally greater consensus and support for TPT in PLHIV and despite this the outcomes do not reach targets.

Lines 227- suggest more discussion of the study findings prior to discussing limitations i.e. would bring other paragraphs (from line 249 onwards) earlier

Lines 224-226- while these findings are important, should clarify that this was not the primary purpose of the review (or was this in the protocol)? Would suggest having some additional context by first citing data from your quantitative findings first to highlight major gaps being TPT being recommended and then completed.

Line 242- again here you are comparing PLHIV to the entire population offered TPT in the earlier review so language should be clear about this as PLHIV are more likely to have additional support services/counseling re: importance of TPT (as you specify in the next sentence) so important to clarify that you are not referring to a comparison between the same populations.

Lines 262-267 - need to acknowledge the limitations (imperfect sensitivity) of these tests in PLHIV given reliance on immune response 

Lines 268-270 - this contradicts what you say in the abstract re: use of LTBI testing did not reduce proportion of people who completed TPT?

Lines 279-284- need to comment on ART drug interactions that may limit use of rifamycins and highlight that clinician training is needed re: safety of shorter regimens with certain ART regimens

Lines 286-287 - Is this really the conclusion the authors want to draw? Average 50% treatment initiation is still not high and below treatment targets. Suggest instead that conclusion should be along the lines of 'TPT initiation and completion in PLHIV fall short of global targets but are higher than observed for other high-risk populations'? We need to be more ambitious as a TB community so drawing the comparison to other data that demonstrates massive unacceptable gaps does not help to drive our ambition to do better.

Table 1

Replace IDU with PWID

Table 4

Re: staff training (enabler) and counseling as time consuming (barrier) - to confirm, this was for TPT initiation but did not pertain to TPT recommendation?

*** Reviewer #3: 

Thank you for the opportunity to read this submission.

1. This piece was a remarkable compilation of data. The analysis of bias was remarkable for the effort but not clear how it eliminated or improved your ability to improve a programs impact. 

2. The inability to understand who was excluded in the studies dilutes your ability to understand the external validity of your findings. 

3. The "care cascade" has its greatest utility at the individual level (single patient), at the clinic site or system level where you can take the findings of your cascade and identify where you need to strengthen your system, i.e, identification, entry and retention. If it is not applied to specific sites it has limited utility as a management tool to direct resources to problems that are contributing to your outcome. As you move into provincial and national data aggregation you dilute the ability to apply corrective interventions along the cascade because your data is aggregated. With aggregated data the ability to locate the barriers to ID, Testing, Initiation TPT, Completion of TPT etc, are precluded and remain disarticulated facts with no place to point the corrective action, to strengthen the cascade. 

4. the Rif vs INH discussion seemed disconnected from data presented and conclusions

*** Reviewer #4: 

In this article the author evaluates the proportion of people living with HIV completing the tuberculosis preventive therapy, assessing the cascade of care, as well as the use of tests to detect the presence of latent tuberculosis infection as part of the cascade and compared similarities in high and low-middle income countries, identifying barriers and facilitators for retention.

This review to be the first specifically looking at the cascade of TPT in PLHIV, and the use of the network meta-analysis methodology is also novel, as prior reviews have conducted more traditional meta-analyses. A major advantage of the network meta-analysis is that this methodology exploits all available direct and indirect evidence. It yields more precise estimates of the intervention effects in comparison with a single direct or indirect estimate and can provide information for comparisons between pairs of interventions that have not been compared directly within an individual randomized trial (indirect comparisons). This method uses the intervention effects from each group of randomized trials and therefore preserves within-trial randomization. Also, the simultaneous comparison of all interventions of interest in the same analysis enables the estimation of their relative ranking for a given outcome. As such, this is an important strength of the manuscript. The manuscript is very well organized and clearly written.

A few points need revision:

1- In abstract, line 54, conclusion: Please rephrase for clarity: "Completion was similar in high and low-middle countries, and in low-middle income countries and the use of LTBI testing did not reduce the proportion of eligible PLHIV who completed TPT." 

2- In order to most properly follow the PRISMA statement, the objective of the systematic review needs further revision, to include the cascade of care participants, interventions, comparisons, and specific outcomes of interest used to identify the articles of interest.

3- In Results, page 160, the sentence "either IGRA or TST (IGRA or TST)" is redundant, please check.

4- Despite results reported in this meta-analysis that by design is restricted to PLHIV are better than those reported for the broader population, it is very important to highlight these results in the context of a still fragmented TB-treatment and prevention service provision, which certainly negatively impact these results, despite the fact that the authors cite that PLHIV are already linked to care.

5- In Table 2, there is a typo ("cohor") that needs correction.In the Discussion, please consider including the strength of evidence for each recommendation or outcome.

6- In the discussion, the authors mention that an important finding in this review was that the use of LTBI tests was not a barrier to TPT initiation or completion among cohorts that used them, and that TPT recommendation and initiation was higher among cohorts that used LTBI tests, and suggest that the use of LTBI tests should be encouraged, not only in high income country where it is already part of care, but also in low and middle-income countries, where this review found numerous reports of its successful use. They also mention that LTBI treatment without prior LTBI testing will expose a substantial proportion of PLHIV to a nontrivial risk of adverse events without evidence of the benefit of TPT. These points need to be put in context considering that published manuscripts not necessarily translate programmatic data. Therefore, taking into account the great utility this review will have from a public health perspective, it is recommended the review of this statement, not to bring the impression that we are currently following guidance that may be harmful to patients, what isn't at all the case. 

7- I suggest to include a paragraph on the most recent trials with shortened course LTBI which will most probably impact the later stages of the cascade, with higher % of finishing the prophylaxis course, although the earlier cascade steps are under the same constrains.

***

[LINK]

---

## [Decision Letter · Decision Letter 2]

11 Jun 2021

Dear Dr. Menzies,

Thank you very much for re-submitting your manuscript "The Latent Tuberculosis Cascade-of-Care Among People Living with HIV: A Systematic Review and Meta-Analysis" (PMEDICINE-D-20-06012R2) for consideration at PLOS Medicine.

I have discussed the paper with editorial colleagues and our academic editor, and it was also seen again by two reviewers. I am pleased to tell you that, provided the remaining editorial and production issues are fully dealt with, we expect to be able to accept the paper for publication in the journal.

[LINK]

Please let me know if you have any questions, and we look forward to receiving the revised manuscript shortly.   

Sincerely,

Richard Turner, PhD

rturner@plos.org

Requests from Editors:

Please resubmit your paper as a research article. 

Please quote the date(s) of the new search in the abstract.

To the abstract, we suggest adding a sentence, say, quoting the proportion of participants in low- and middle-income countries, and broken down by WHO region. 

We also suggest adding a sentence to summarize participants' age, sex, and CD4 and ART status. 

In the abstract and throughout the text, please quote p values alongside 95% CI, where available. 

Please remove the information about study funding from the abstract. In the event of publication, this information will appear in the article metadata, via entries in the submission form. 

At lines 80/81, we suggest quoting both numbers to one decimal place. 

At line 221, please make that "1.8 million".

Please use the style "low- and middle-income" throughout.

Please use the journal name abbreviation "PLoS ONE" consistently in the reference list.

Are references 14 and 57-65 missing journal names?

Please rename the attached PRISMA checklist "S1_PRISMA_Checklist" or similar and refer to it by this label at line 116.

Comments from Reviewers:

*** Reviewer #1: 

The authors have addressed all my points.

Michael Dewey

*** Reviewer #2: 

PMEDICINE-D-20-06012_R2

The authors have responded thoughtfully and appropriately to the critiques I and the other reviewers raised and accordingly present a strengthened manuscript, which includes an updated search and additional studies as well as revisions that have improved the clarity and messaging of the manuscript. I thus do not have major recommendations to offer at this stage. I would however suggest that the authors consider adding text in the abstract and author summary conclusions that note that TPT recommendation and initiation was higher among cohorts that used LTBI tests. For example, they could add something to the effect of the following clause (in bold) in the author's conclusion: 'Although TPT recommendation and initiation was higher among cohorts that used LTBI tests, the use of LTBI testing did not reduce the proportion of eligible PLHIV who completed TPT.' The rationale for this suggestion is that this paper may have important policy implications and it is otherwise possible that it could be interpreted that LTBI testing should not be scaled up in LMICs, which is not in accordance with the author's suggested recommendations. Otherwise, the manuscript needs a careful review by the authors to correct a few minor typos but overall represents an important addition to the literature on this topic.

***

[LINK]

---

## [Editor Report · Decision Letter 3]

20 Jun 2021

Dear Dr Menzies, 

On behalf of my colleagues and our Academic Editor, I am pleased to inform you that we have agreed to publish your manuscript "The Latent Tuberculosis Cascade-of-Care Among People Living with HIV: A Systematic Review and Meta-Analysis" (PMEDICINE-D-20-06012R3) in PLOS Medicine.

Prior to final acceptance, please check numbers for consistency throughout the paper - we think that the proportion of participants on ART is quoted as 46.3% in the abstract and 44% early in the Results section (the denominators appear to be different). 

PRESS

Sincerely, 

Richard Turner, PhD 

rturner@plos.org